# Assignment of Absolute Configuration of Bromoallenes by Vacuum-Ultraviolet Circular Dichroism (VUVCD)

**DOI:** 10.3390/molecules26051296

**Published:** 2021-02-27

**Authors:** Taiki Umezawa, Nakaba Mizutani, Koichi Matsuo, Yuugo Tokunaga, Fuyuhiko Matsuda, Tatsuo Nehira

**Affiliations:** 1Division of Environmental Materials Science, Graduate School of Environmental Science, Hokkaido University, N10W5, Kita-ku, Sapporo 060-0810, Japan; gingnangboyzontherun@gmail.com (N.M.); fmatsuda@ees.hokudai.ac.jp (F.M.); 2Hiroshima Synchrotron Radiation Center (HiSOR), Hiroshima University, 2-313 Kagamiyama, Higashi-Hiroshima 739-0046, Japan; pika@hiroshima-u.ac.jp; 3Faculty of Integrated Arts and Sciences, Hiroshima University, 1-7-1 Kagamiyama, Higashi-Hiroshima 739-8521, Japan; yugotokunaga@aol.com; 4Graduate School of Integrated Sciences for Life, Hiroshima University, 1-7-1 Kagamiyama, Higashi-Hiroshima 739-8521, Japan

**Keywords:** bromoallene, structure determination, VUVCD, Lowe’s rule, natural product

## Abstract

A new application of vacuum-ultraviolet circular dichroism (VUVCD), which enables the measurement of CD spectra in the vacuum-ultraviolet region (140–200 nm), for the assignment of the absolute configurations of bromoallenes is described. Bromoallene moieties are found in natural products obtained from many marine organisms. To date, the absolute configuration of bromoallenes has been assigned almost exclusively with Lowe’s rule, which is based on specific rotation. However, exceptions to Lowe’s rule have been reported arising from the presence of other substituents with large specific rotations. For the unambiguous assignment of the absolute configuration of the bromoallene moiety with its characteristic absorption wavelength at 180–190 nm due to the π–π* transition, VUVCD was applied to four pairs of bromoallene diastereomers prepared by modifying the synthetic scheme of omaezallene. The VUVCD spectra clearly showed positive or negative Cotton effects around 180–190 nm according to the configuration of the bromoallene employed, revealing the potential of VUVCD for determining absolute stereochemistry.

## 1. Introduction

Natural products with bromoallene moieties are found in many marine organisms and possess various biological activities [1]. 1,3-Disubstituted bromoallene has an axial chirality that induces polarization. In general, the specific rotation by the bromoallene functionality shows a large absolute value whose sign corresponds to its absolute configuration. This is based on Lowe’s rule, an empirical rule in which a minus value is assigned to a bromoallene with *R* configuration and a plus value to the *S* configuration [2]. Lowe’s rule has been widely employed for the assignment of absolute configurations of allene moieties found in newly isolated natural products [3,4,5,6,7,8,9,10,11,12]. However, exceptions to Lowe’s rule have been reported. Specific rotations of allene moieties can be disrupted by the influence of other substituents [13,14]. We encountered an exception to Lowe’s rule in the total synthesis of omaezallene (**1**) (Figure 1) [13]. Although **1** and *Z*-omaezallene (**2**) followed the rule, their 9-epimers, 9-*epi*-**1** and 9-*epi*-**2**, had opposite signs of specific rotation to those of **1** and **2**. This is presumably attributable to interference from the substituent at the 9 position. Thus, for the unambiguous assignment of the absolute configuration of the bromoallene moiety, development of a convenient and reliable methodology was required.

We envisioned vacuum-ultraviolet circular dichroism (VUVCD), which measures CD spectra in the vacuum-ultraviolet region (minimum wavelength is 140 nm) [15,16], for unambiguous determination of absolute configuration. The bromoallene moiety typically has a characteristic absorption around 180–190 nm corresponding to the π–π* transition [17]. We anticipated that expanding the utility of VUVCD would be helpful since VUVCD has provided extra insights into the absolute configurations and conformations of biomolecules [18], such as sugars [19], amino acids [20], and proteins [21,22]. Herein, we describe syntheses of a series of bromoallene diastereomers by modifying the synthetic scheme of omaezallene and their VUVCD spectral analyses.

## 2. Results and Discussion

In order to prepare object-oriented diastereomeric pairs that feature both configurations at the bromoallene moiety, modifications to our previous synthesis were necessary. Construction of propargyl alcohol **5**, a stereospecific precursor for the bromoallene, was previously carried out in a stereoselective manner with acetylide and hemiacetal **4** obtained from d-glucose through aldehyde **3** (Scheme 1). We believe this high stereoselectivity was due to the hydroxy group in the α-hydroxyaldehyde **4′**, which was in equilibrium with hemiacetal **4** [13]. Thus, we planned the direct addition of the acetylide to aldehyde **3**, anticipating that the absence of the hydroxy group in the substrate would provide a mixture of diastereomeric alcohols **6** and **7**. Further manipulations of the acetal moiety were expected to give diastereomeric alcohols **8** as appropriate intermediates for both enantiomers of omaezallene.

As shown in Scheme 2, the synthesis began with aldehyde **3** [23], which was treated with acetylide to give diastereomeric propargyl alcohols **6** and **7** as an inseparable mixture (dr = 1.6:1). After several attempts to separate the diastereomers, esterification of the mixture with a commercially available (*R*)-*O*-methylmandelic acid allowed separation of the two isomers by silica gel column chromatography. The absolute configurations of the propargyl alcohols were revealed to be *R* for the less polar ester **9** and *S* for the more polar ester **10** by the modified Mosher method [24] (see Appendix A in the Supplementary Material). Thus, both isomers of the bromoallene were now accessible. Ester **10** was converted to bromoallene (1*S*)-**12** in 3 steps: hydrolysis of the ester, sulfonate formation [25,26], and stereospecific installation of the bromoallene moiety to sulfonate **11**. A two-step sequence through deprotection of the acetal with TsOH, followed by Wittig reaction [27], afforded unsaturated ester (1*S*)-**13**. After reduction of the ester to an allylic alcohol, tetrahydrofuran (1*S*)-**14** was successfully obtained by bromoetherification with NBS in a highly diastereoselective manner [28,29,30,31] without loss of the bromoallene functionality. The treatment of (1*S*)-**14** with TBSOTf and 2,6-lutidine provided silyl ether (1*S*)-**8** after the less hindered TBS ether was selectively removed under mild acidic conditions [32]. ^1^H and ^13^C NMR spectra of (1*S*)-**8** were identical with those of the previous intermediate of omaezallene [13]. Furthermore, the specific rotation of this newly prepared (1*S*)-**8** was +61.3 (c 1.10, CHCl_3_), indicating that the enantiomer of the previous intermediate [−68.0 (c 0.75, CHCl_3_)] was successfully prepared from the same starting material [13]. Diastereomer **9** was converted to alcohol (1*R*)-**8** by the same scheme as for (1*S*)-**8**.

With several pairs of bromoallene diastereomers in hand, (1*S*)-/(1*R*)-**12**, (1*S*)-/(1*R*)-**13**, (1*S*)-/(1*R*)-**14**, and (1*S*)-/(1*R*)-**15**, we turned our attention to collecting VUVCD spectra. The results are summarized in Figure 2. These compounds followed Lowe’s rule for bromoallene structures, exhibiting positive and negative values of specific rotations for the *S* and *R* isomers, respectively. All compounds had the characteristic absorption around 190 nm corresponding to the π–π* transition in UV spectroscopy. Positive Cotton effects were observed at the wavelength region of interest for all *S* bromoallenes and negative ones for all *R* bromoallenes, despite the presence of various functional groups such as hydroxy groups, acetals, esters, or silyl ethers. The relationship between the signs of the CD spectra of all of these compounds and their absolute configurations is consistent well with that obtained for other chiral allenes [33], implying that this π–π* transition of interest may be applicable to other types of allene derivatives. While these transitions were previously either non-observable or visualized only by gas-phase measurements, our results demonstrate the potential of VUVCD for the robust assignment of absolute configurations of bromoallenes.

The obtained VUVCD spectra of all the bromoallenes were compared to the theoretical CD spectra predicted by quantum mechanics calculations [34]. Starting from a conformational search, the resultant stable conformers were optimized by density functional theory calculations at the B3LYP/6-31G(d) level of theory, followed by conformational analysis by Boltzmann’s law to estimate dominant conformers. Theoretical CD spectra were produced by performing time-dependent density functional theory calculations at the double-zeta level using several functionals that had been previously used in several CD calculation studies. Although spectra from different functionals did not always agree well with the experimental ones as shown in Figure 3, comparison between several functionals provided us with an encouraging sense of accomplishment: the conclusion was always the same no matter what functional was used. In the current study, the functional B3LYP tended to give better agreement than BHandHLYP, or CAM-B3LYP. Changing basis sets did not significantly affect the shape of the theoretical spectra.

This approach of VUVCD measurement and interpretation has yielded the following three important arguments for this method. First, bromoallenes were expected to show intense Cotton effects at the wavelength corresponding to distinct absorption peaks at 180–190 nm, which were clearly observable by VUVCD and agreed well with the previous report [33]. This is also encouraging since a couple of reports demonstrated that VUVCD spectra were applicable to solutions of not only water but also organic solvents [35,36]. Second, the signs of the Cotton effect in the bromoallenes were in accordance with the absolute configurations in both experimental VUVCD and theoretical CD spectra throughout the models of this study. This configurational discussion showed a complete match with the reported work of our omaezallene synthesis [13]. Third, both the absorption peak and the CD extreme of the bromoallenes were easy to assign because they were isolated from peaks arising from other functional groups. Since this theoretical prediction of CD spectra is nonempirical, requiring no reference sample, the method is expected to be helpful for any sample of interest as long as the experiment is feasible.

In summary, concise preparations of bromoallene diastereomers and their VUVCD analyses have been described. Addition of an acetylide Grignard reagent to an aldehyde gave a mixture of diastereomers, which were separated by esterification with an optically active carboxylic acid. Further transformations enabled the synthesis of diastereomeric pairs of various bromoallene-containing omaezallene intermediates. The current VUVCD results demonstrate its potential for the unambiguous assignment of the absolute configuration of bromoallenes instead of Lowe’s rule. It is also advantageous that VUVCD spectra of bromoallenes can be interpreted theoretically. Further research on various bromoallenes, including compounds that do not follow Lowe’s rule, are underway.

## 3. Materials and Methods

### 3.1. General Methods

Tetrahydrofuran, diethyl ether, acetone, acetonitrile, and 1,2-dichloroethane (dehydrated grade) were purchased from Kanto Chemical Co., (Tokyo, Japan). Inc. Dichloromethane (CH_2_Cl_2_), and triethylamine (Et_3_N) were distilled from CaH_2_. All commercially obtained reagents were used as received. Analytical and preparative TLC were carried out using pre-coated silica gel plates (Macherey-Nagel DC-Fertigplatten SIL G-25 UV_254_), purchased from Merck (Darmstadt, Germany). Merck Kieselgel 60 Art 7734 silica gel was used for column chromatography. IR spectra were recorded on a JASCO FTIR-4100 Type A spectrometer, purchased from JASCO (Tokyo, Japan), using a NaCl cell or a KBr board. ^1^H- and ^13^C-NMR spectra were recorded using a JNM-EX 400 (400 MHz and 100 MHz) spectrometer, purchased from JEOL (Tokyo, Japan). Chemical shifts are reported in ppm relative to CHCl_3_ (δ = 7.26) in CDCl_3_ for ^1^H NMR, and CDCl_3_ (δ = 77.0) for ^13^C NMR. Splitting patterns are designated as s, d, t, q, and m, indicating singlet, doublet, triplet, quartet, and multiplet, respectively. The VUVCD spectra were measured in the wavelength region from 200 to 175 nm at 25 °C using a VUVCD spectrophotometer at the Hiroshima Synchrotron Radiation Center (HiSOR), Hiroshima University [18]. This spectrophotometer uses a synchrotron radiation light source to observe the CD spectrum in the high energy region. The details of the optical devices of the spectrophotometer are available elsewhere [15,16]. The VUVCD measurements were carried out using an assembled-type optical cell with CaF_2_ windows [18]. The path length in the cell was adjusted using a Teflon spacer to 50 μm. The sample solutions were freshly prepared by dissolving in acetonitrile at an appropriate concentration, which was determined based on the reported molar absorption coefficient [33]. All of the spectra were recorded using a 1.0-mm slit, 4-s time constant, 20-nm/min scan speed, and four accumulations. UV absorbance was derived from the high-tension voltage for the CD measurement. Conformational searches were performed with CONFLEX 8 (Version 8.A.0901 by CONFLEX, Tokyo) [37] using a commercially available PC (Operating System: Windows 10 Pro for Workstations, CPU: Intel Xeon E5-1650 v4 processor 3.60 GHz, RAM 32 GB) and density functional theory (DFT) calculations were conducted with Gaussian 09 (Revision E.01 by Gaussian, Wallingford, CT) [38] with a PC (Operating System: CentOS 7 a Linux, CPU: Intel Xeon E5-1660 v4 processor 3.20 GHz, RAM 32 GB).

### 3.2. Syntheses

#### 3.2.1. Propargyl Alcohols **6** and **7**

To a solution of aldehyde **3** (3.10 g, 18.0 mmol) in THF (50 mL) was added 0.5 M ethynylmagnesium chloride (72.0 mL, 36.0 mmol) at 0 °C under Ar atmosphere. The mixture was stirred for 2 h at 0 °C, quenched with diluted HCl, extracted with AcOEt, washed with brine, dried over Na_2_SO_4_, filtered, and concentrated in vacuo. The crude product was purified by silica gel column chromatography (AcOEt:hexane = 15:85 then 30:70) to give mixture of propargyl alcohols **6** and **7** (3.07 g, 15.5 mmol, 86%) as an inseparable mixture:

Spectroscopic data of **6** and **7** after separation by next conversions are follows:

**6** [α]_D_^24^ −14.9 (c 1.07, CHCl_3_); IR (neat) 3448, 3273, 2989, 2937, 2904, 2114, 1456, 1435, 1375, 1258, 1219, 1164, 1056, 1024, 850, 665, 540, 524 cm^–1^; ^1^H NMR (400 MHz, CDCl_3_) δ 1.33 (3H, s), 1.53 (3H, s), 1.84 (1H, ddd, *J* = 14.4, 9.5, 4.9 Hz), 2.19 (1H, dd, *J* = 13.7, 4.4 Hz), 2.28 (1H, d, *J* = 6.3 Hz), 2.50 (1H, d, *J* = 2.0 Hz), 4.31–4.39 (2H, m), 4.77 (1H, t, *J* = 4.1 Hz), 5.84 (1H, dd, *J* = 2.0 Hz); ^13^C NMR (CDCl_3_, 100 MHz) δ 26.2, 26.8, 34.7, 64.0, 74.2, 80.5, 80.6, 81.2, 105.9, 111.7; HRMS (ESI) *m*/*z*: [M + Na]^+^; calculated for C_10_H_14_O_4_Na 221.0788; Found 221.0787.

**7** [α]_D_^23^ −15.6 (c 0.96, CHCl_3_); IR (neat) 3473, 3254, 2994, 2984, 2965, 2948, 2904, 2120, 1435, 1372, 1321, 1255, 1212, 1167, 1123, 1053, 1018, 938, 850, 720, 673 cm^–1^; ^1^H NMR (400 MHz, CDCl_3_) δ 1.33 (3H, s), 1.53 (3H, s), 2.03 (1H, ddd, *J* = 14.4, 10.2, 4.4 Hz), 2.16 (1H, dd, *J* = 13.7, 4.9 Hz), 2.24 (1H, d, *J* = 4.9 Hz), 2.46 (1H, d, *J* = 2.9 Hz), 4.42 (1H, ddd, *J* = 10.4, 4.9, 3.4 Hz), 4.63 (1H, quin, *J* = 2.6 Hz), 4.78 (1H, t, *J* = 4.4 Hz), 5.85 (1H, dd, *J* = 3.9 Hz); ^13^C NMR (CDCl_3_, 100 MHz) δ 26.2, 26.8, 32.5, 62.3, 74.4, 80.2, 80.5, 105.8, 111.7; HRMS (ESI-orbitrap) *m*/*z*: [M + Na]^+^; calculated for C_10_H_14_O_4_Na 221.0788; Found 221.0784.

#### 3.2.2. Ester **9** and **10**

To a solution of **6** and **7** (mixture, 713 mg, 3.59 mmol) in CH_2_Cl_2_ (20 mL) were added (*R*)-*O*-methylmandelic acid (628 mg, 3.77 mmol), DMAP (43.9 mg, 0.359 mmol) and EDCI (897 mg, 4.67 mmol) at room temperature under Ar atmosphere. The mixture was stirred at room temperature for 1.5 h, quenched with saturated NaHCO_3_, extracted with AcOEt, washed with brine, dried over Na_2_SO_4_, filtered, and concentrated in vacuo. The crude product was purified by silica gel column chromatography (AcOEt:hexane = 7:93) to give ester **10** (671 mg, 1.94 mmol, 54%) as a colorless oil and ester **9** (419 mg, 1.21 mmol, 34%) as a colorless oil:

**9** [α]_D_^23^ −74.7 (c 0.87, CHCl_3_); IR (neat) 3269, 2989, 2938, 2899, 2830, 2127, 1758, 1452, 1370, 1317, 1213, 1166, 1116, 1076, 1025, 848, 743, 700 cm^–1^; ^1^H NMR (400 MHz, CDCl_3_) δ 1.24 (3H, s), 1.41 (3H, s), 1.66 (1H, ddd, *J* = 13.7, 10.0, 4.9 Hz), 2.00 (1H, dd, *J* = 13.7, 4.9 Hz), 2.48 (1H, d, *J* = 2.4 Hz), 3.39 (3H, s), 4.29 (1H, dt, *J* = 9.3, 4.1 Hz), 4.40 (1H, t, *J* = 4.2 Hz), 4.81 (1H, s), 5.08 (1H, d, *J* = 3.4 Hz), 5.74 (1H, t, *J* = 2.7 Hz), 7.33–7.40 (3H, m), 7.42–7.45 (2H, m); ^13^C NMR (CDCl_3_, 100 MHz) δ 25.9, 26.5, 32.3, 57.0, 62.9, 75.0, 78.4, 79.6, 81.7, 105.3, 111.3, 127.1, 128.3, 128.4, 135.8, 168.5; HRMS (ESI-orbitrap) *m*/*z*: [M + Na]^+^; calculated for C_19_H_22_O_6_Na 369.1309; Found 369.1314.

**10** [α]_D_^23^ −24.1 (c 1.68, CHCl_3_); IR (neat) 3266, 3066, 3034, 2989, 2938, 2904, 2830, 2124, 1758, 1494, 1452, 1373, 1320, 1238, 1213, 1166, 1105, 1076, 1054, 1025, 957, 883, 848, 737, 698 cm^–1^; ^1^H NMR (400 MHz, CDCl_3_) δ 1.25 (3H, s), 1.52 (3H, s), 1.76 (1H, ddd, *J* = 13.9, 10.2, 4.9 Hz), 2.20 (1H, dd, *J* = 13.7, 4.9 Hz), 2.35 (1H, d, *J* = 2.0 Hz), 3.46 (3H, s), 4.42 (1H, ddd, *J* = 10.6, 6.6, 4.9 Hz), 4.72 (1H, t, *J* = 4.2 Hz), 4.84 (1H, s), 5.46 (1H, dd, *J* = 6.6, 2.0 Hz), 5.82 (1H, d, *J* = 3.9 Hz), 7.33–7.40 (3H, m), 7.43–7.46 (2H, m); ^13^C NMR (CDCl_3_, 100 MHz) δ 26.4, 27.0, 35.4, 57.5, 65.9, 75.8, 78.0, 80.2, 82.3, 106.2, 111.9, 127.6, 128.8, 128.9, 135.7, 169.5; HRMS (ESI-orbitrap) *m*/*z:* [M + Na]^+^; calculated for C_19_H_22_O_6_Na 369.1309; Found 369.1314.

#### 3.2.3. Sulfonate **11**

To a solution of **10** (505 mg, 1.46 mmol) in THF (5.0 mL) was added 1.0 M NaOH solution (5.0 mL) at room temperature. The mixture was stirred for 1 h, extracted with Et_2_O, washed with brine, dried over Na_2_SO_4_, filtered, and concentrated in vacuo. Crude propargyl alcohol was directly employed in the next reaction.

To a solution of the crude propargyl alcohol in CH_2_Cl_2_ (5.0 mL) were added DMAP (366 mg, 3.00 mmol) and TrisCl (848 mg, 2.80 mmol) at room temperature under Ar atmosphere. The mixture was stirred for 12 h and concentrated in vacuo. The residue was purified by silica gel column chromatography (AcOEt:hexane = 2:98 then 8:92) to give sulfonate **11** (612 mg, 1.32 mmol, 91%) as a colorless oil: [α]_D_^23^ +5.5 (c 1.19, CHCl_3_); IR (neat) 3280, 2960, 2870, 2127, 1600, 1563, 1462, 1428, 1380, 1351, 1216, 1179, 1105, 1028, 960, 846, 808, 666 cm^–1^; ^1^H NMR (400 MHz, CDCl_3_) δ 1.24 (3H, s), 1.26 (6H, d, *J* = 2.4 Hz), 1.27 (12H, d, *J* = 2.4 Hz), 1.48 (3H, s), 1.96 (1H, ddd, *J* = 14.3, 9.5, 4.9 Hz), 2.22 (1H, dd, *J* = 13.7, 4.9 Hz), 2.34 (1H, d, *J* = 2.4 Hz), 2.90 (1H, sep, *J* = 7.0 Hz), 4.12 (2H, sep, *J* = 6.7 Hz), 4.45 (1H, quin, *J* =5.0 Hz), 4.74 (1H, t, *J* = 4.1 Hz), 5.24 (1H, dd, *J* = 5.1, 2.4 Hz), 5.80 (1H, d, *J* = 3.9 Hz), 7.16 (2H, s); ^13^C NMR (CDCl_3_, 100 MHz) δ 23.5, 24.5, 24.6, 24.7, 26.2, 26.9, 29.6, 31.5, 34.2, 69.8, 76.2, 76.6, 77.0, 77.2, 77.3, 77.4, 77.9, 78.7, 80.1, 105.9, 111.9, 123.6, 130.2, 150.7, 153.9; HRMS (ESI-orbitrap) *m*/*z*: [M + Na]^+^; Calcd for C_25_H_36_O_6_NaS 487.2136; Found 487.2134.

#### 3.2.4. Bromoallene (1*S*)-**12**

To a solution of **11** (482 mg, 1.04 mmol) in THF (6.0 mL) were added LiBr (903 mg, 10.4 mmol) and CuBr (1.49 g, 10.4 mmol) at room temperature under Ar atmosphere. The mixture was stirred at 60 °C for 2 h, quenched with saturated NaHCO_3_, filtered through celite pad, extracted with AcOEt, washed with brine, dried over Na_2_SO_4_, filtered through short silica gel pad, and concentrated in vacuo. The crude product was purified by silica gel column chromatography (AcOEt:hexane = 5:95) to give bromoallene (1*S*)-**12** (154 mg, 0.590 mmol, 57%) as a colorless oil: [α]_D_^23^ +105.7 (c 0.38, CHCl_3_); IR (neat) 3735, 3650, 2931, 1375, 1015, 850, 660 cm^–1^; ^1^H NMR (400 MHz, CDCl_3_) δ 1.33 (3H, s), 1.52 (3H, s), 1.71 (1H, ddd, *J* = 13.4, 10.7, 4.4 Hz), 2.27 (1H, dd, *J* = 13.7, 4.4 Hz), 4.75–4.82 (3H, m), 5.44 (1H, t, *J* = 6.1 Hz), 5.85 (1H, d, *J* = 3.9 Hz), 6.10 (1H, dd, *J* = 5.4, 1.5 Hz); ^13^C NMR (CDCl_3_, 100 MHz) δ 25.8, 26.4, 38.8, 73.6, 74.2, 80.0, 99.2, 105.2, 111.1, 202.0; HRMS (ESI-orbitrap) *m*/*z*: [M + H]^+^; calculated for C_10_H_14_O_3_^79^Br 261.0121; Found 261.0123.

#### 3.2.5. Unsaturated Ester (1*S*)-**13**

To a solution of (1*S*)-**12** (334 mg, 1.28 mmol) in CH_3_CN-H_2_O (11 mL, 10/1) was added TsOH·H_2_O (33.9 mg) at room temperature under Ar atmosphere. The mixture was stirred at 60 °C for 6 h, quenched with saturated NaHCO_3_, extracted with AcOEt, washed with brine, dried over Na_2_SO_4_, filtered, and concentrated in vacuo. Crude hemiacetal was directly employed in the next reaction.

To a solution of the crude hemiacetal in CH_2_Cl_2_ (10 mL) was added the *n*Bu_3_P=CHCO_2_Et (1.00 M in toluene, 2.67 mL, 2.67 mmol) at room temperature under Ar atmosphere. The mixture was stirred for 1 h and concentrated in vacuo. The crude product was purified by silica gel column chromatography (AcOEt:hexane = 20:80) to give unsaturated ester (1*S*)-**13** (80.4 mg, 0.277 mmol, 31% over 2 steps) as a colorless oil: [α]_D_^23^ +31.8 (c 0.68, CHCl_3_); IR (neat) 3854, 3732, 2979, 1698, 1540, 1308, 1187, 1040 cm^–1^; ^1^H NMR (400 MHz, CDCl_3_) δ 1.31 (3H, t, *J* = 7.1 Hz), 1.76–1.92 (2H, m), 4.20 (2H, q, *J* = 7.2 Hz), 4.58–4.63 (2H, m), 5.47 (1H, t, *J* = 5.9 Hz), 6.10 (1H, dd, *J* = 15.6, 1.5 Hz), 6.14 (1H, dd, *J* = 5.9, 2.0 Hz), 6.92 (1H, dd, *J* = 15.6, 4.9 Hz); ^13^C NMR (CDCl_3_, 100 MHz) δ 14.5, 42.4, 60.9, 69.3, 71.0, 75.3, 103.8, 120.9, 148.9, 166.7, 200.9; HRMS (ESI-orbitrap) *m*/*z*: [M + Na]^+^; calculated for C_11_H_15_O_4_^79^BrNa 313.0046; Found 313.0045.

#### 3.2.6. Bromoether (1*S*)-**14**

To a solution of (1*S*)-**13** (58.9 mg, 0.200 mmol) in CH_2_Cl_2_ (1.0 mL) was added DIBAL (1.00 M in toluene, 1.00 mL, 1.00 mmol) at 0 °C under Ar atmosphere. The mixture was stirred for 30 min, quenched with 1.00 M HCl, stirred for 30 min, extracted with AcOEt, washed with brine, dried over Na_2_SO_4_, filtered, and concentrated in vacuo to give crude alcohol, which was directly employed in the next reaction.

To a solution of the crude alcohol in CH_3_CN (1.0 mL) was added NBS (35.6 mg, 0.200 mmol) at 0 °C under Ar atmosphere. The mixture was stirred for 30 min, quenched with 20% Na_2_S_2_O_3_ aqueous solution, extracted with AcOEt, washed with brine, dried over Na_2_SO_4_, filtered, and concentrated in vacuo. The crude product was purified by silica gel column chromatography (AcOEt:hexane = 30:70) to give bromoether (1*S*)-**14** (39.2 mg, 0.120 mmol, 60% over 2 steps) as a colorless oil: [α]_D_^23^ +83.5 (c 0.52, CHCl_3_); IR (neat) 3851, 3735, 3649, 2931, 1698, 1507, 1034, 670 cm^–1^; ^1^H NMR (400 MHz, CDCl_3_) δ 2.03 (1H, ddd, *J* = 13.8, 9.3, 4.4 Hz), 2.27 (1H, dd, *J* = 13.4, 5.9 Hz), 3.97 (1H, dd, *J* = 12.2, 3.9 Hz), 4.02 (1H, dd, *J* = 12.2, 4.4 Hz), 4.17–4.22 (2H, m), 4.58–4.64 (1H, m), 4.90–4.97 (1H, m), 5.44 (1H, t, *J* = 5.9 Hz), 6.10 (1H, dd, *J* = 5.9, 1.5 Hz); ^13^C NMR (CDCl_3_, 100 MHz) δ 40.5, 51.3, 65.7, 73.1, 74.1, 76.2, 84.1, 101.1, 201.7; HRMS (ESI-orbitrap) *m*/*z*: [M + Na]^+^; calculated for C_9_H_12_O_3_^79^Br_2_Na 348.9045; Found 348.9053.

#### 3.2.7. TBS Ether (1*S*)-**15**

To a solution of (1*S*)-14 (61.5 mg, 0.190 mmol) in CH_2_Cl_2_ (1.0 mL) were added 2,6-lutidine (110 μL, 0.950 mmol) and TBSOTf (131 μL, 0.570 mmol) at 0 °C under an Ar atmosphere. The mixture was stirred at room temperature for 2 h, quenched with saturated NaHCO_3_, extracted with AcOEt, washed with 1.00 M HCl and brine, dried over Na_2_SO_4_, filtered, and concentrated in vacuo. Crude TBS ether (1*S*)-15 was sufficiently pure without column chromatography: [α]_D_^23^ +49.6 (c 0.81, CHCl_3_); IR (neat) 3851, 3735, 2928, 1700, 1540, 1254, 1046, 836, 668 cm^–1^; ^1^H NMR (400 MHz, CDCl_3_) δ 0.08 (3H, s), 0.11 (3H, s), 0.12 (3H, s), 0.16 (3H, s), 0.91 (18H, s), 1.90 (1H, ddd, *J* = 13.2, 9.8, 3.4 Hz), 2.12 (1H, dd, *J* = 13.2, 5.9 Hz), 3.95–4.15 (4H, m), 4.50 (1H, s), 4.78–4.87 (1H, m), 5.42 (1H, t, *J* = 6.1 Hz), 6.06 (1H, dd, *J* = 5.6, 2.0 Hz); ^13^C NMR (CDCl_3_, 100 MHz) δ −5.3, −4.8, −4.3, −0.08, 18.0, 18.4, 25.8, 25.9, 41.7, 52.5, 65.1, 73.4, 73.7, 73.8, 75.4, 75.7, 77.2, 82.9, 83.0, 102.0, 201.4; HRMS (ESI-orbitrap) *m*/*z*: [M + Na]^+^; calculated for C_21_H_40_O_3_^79^Br_2_NaSi_2_ 577.0775; Found 577.0778.

#### 3.2.8. Alcohol (1*S*)-**8**

To a solution of the crude(1*S*)-**15** in CH_2_Cl_2_-MeOH (0.2 mL, 2:1) was added CSA (13.2 mg, 0.0570 mmol) at room temperature under Ar atmosphere. The mixture was stirred for 90 min, quenched with Et_3_N (7.9 μL, 0.0570 mmol), and concentrated in vacuo. The crude product was purified by silica gel column chromatography (AcOEt:hexane = 30:70) to give alcohol (1*S*)-**8** (22.5 mg, 0.0513 mmol, 27% over 2 steps) as a colorless oil: [α]_D_^23^ +61.3 (c 1.10, CHCl_3_); IR (neat) 3447, 3057, 2953, 2929, 2884, 2856, 2362, 2362, 1959, 1471, 1437, 1361, 1255, 1177, 1106, 1044, 1006, 936, 836, 807, 776, 709, 660 cm^–1^; ^1^H NMR (400 MHz, CDCl_3_) δ 0.13 (3H, s), 0.17 (3H, s), 0.92 (9H, s), 1.90 (1H, ddd, *J* = 13.2, 9.8, 3.7 Hz), 2.13 (1H, dd, *J* = 12.9, 5.6 Hz), 3.94–4.02 (2H, m), 4.13–4.21 (2H, m), 4.52 (1H, t, *J* = 2.9 Hz), 4.84–4.89 (1H, m), 5.43 (1H, t, *J* = 5.6 Hz), 6.09 (1H, t, *J* = 5.6 Hz); ^13^C NMR (CDCl_3_, 100 MHz) δ −4.8, −4.2, 18.0, 25.9, 41.5, 41.6, 50.6, 66.0, 73.2, 74.0, 75.7, 76.1, 85.4, 101.4, 201.4; HRMS (ESI-orbitrap) *m*/*z*: [M + Na]^+^; calculated for C_15_H_26_O_3_^79^Br_2_NaSi 462.9910; Found 462.9914.

#### 3.2.9. Sulfonate **S1** (the Corresponding Diastereomer of **11**)

To a solution of **9** (395 mg, 1.14 mmol) in THF (5.0 mL) was added 1.0 M NaOH solution (5.0 mL) at room temperature. The mixture was stirred for 1 h, extracted with Et_2_O, washed with brine, dried over Na_2_SO_4_, filtered, and concentrated in vacuo. Crude propargyl alcohol was directly employed in the next reaction.

To a solution of the crude propargyl alcohol in CH_2_Cl_2_ (5.0 mL) were added DMAP (366 mg, 3.00 mmol) and TrisCl (848 mg, 2.80 mmol) at room temperature under Ar atmosphere. The mixture was stirred for 12 h and concentrated in vacuo. The residue was purified by silica gel column chromatography (AcOEt:hexane = 2:98 then 8:92) to give sulfonate **S1** (465 mg, 1.00 mmol, 88%) as a colorless oil: [α]_D_^23^ −18.5 (c 1.08, CHCl_3_); IR (neat) 3279, 2957, 2936, 2875, 2128, 1601, 1565, 1463, 1428, 1381, 1350, 1182, 1033, 895, 671 cm^–1^; ^1^H NMR (400 MHz, CDCl_3_) δ 1.24 (3H, s), 1.25 (6H, d, *J* = 4.9 Hz), 1.27 (12H, d, *J* = 4.4 Hz), 1.48 (3H, s), 1.95 (1H, ddd, *J* = 14.4, 9.3, 4.9 Hz), 2.21 (1H, dd, *J* = 13.7, 4.9 Hz), 2.35 (1H, d, *J* = 2.0 Hz), 2.90 (1H, sep, *J* = 7.0 Hz), 4.13 (2H, sep, *J* = 6.7 Hz), 4.45 (1H, ddd, *J* =10.1, 4.9, 3.9 Hz), 4.71 (1H, t, *J* = 4.1 Hz), 5.32 (1H, dd, *J* = 3.4, 2.4 Hz), 5.73 (1H, t, *J* = 3.9 Hz), 7.16 (2H, s); ^13^C NMR (CDCl_3_, 100 MHz) δ 23.5, 24.5, 24.6, 24.7, 26.2, 26.9, 29.6, 33.7, 34.2, 69.5, 76.1, 78.7, 80.0, 106.0, 111.7, 123.6, 130.4, 150.7, 153.8; HRMS (ESI-orbitrap) *m*/*z*: [M + Na]^+^; Calcd for C_23_H_36_O_6_NaS 487.2125; Found 487.2121.

#### 3.2.10. Bromoallene (1*R*)-**12**

To a solution of **S1** (321 mg, 0.690 mmol) in THF (5.0 mL) were added LiBr (599 mg, 6.90 mmol) and CuBr (990 mg, 6.90 mmol) at room temperature under Ar atmosphere. The mixture was stirred at 60 °C for 2 h, quenched with saturated NaHCO_3_, filtered through celite pad, extracted with AcOEt, washed with brine, dried over Na_2_SO_4_, filtered through short silica gel pad, and concentrated in vacuo. The crude product was purified by silica gel column chromatography (AcOEt:hexane = 5:95) to give unsaturated bromoallene (1*R*)-**12** (85.0 mg, 0.330 mmol, 47%) as a colorless oil: [α]_D_^23^ −101.7 (c 0.69, CHCl_3_); IR (neat) 3735, 2934, 1375, 1212, 1015, 850, 660 cm^–1^; ^1^H NMR (400 MHz, CDCl_3_) δ 1.33 (3H, s), 1.53 (3H, s), 1.71 (1H, ddd, *J* = 13.4, 10.7, 4.4 Hz), 2.24 (1H, dd, *J* = 13.7, 4.4 Hz), 4.75–4.81 (2H, m), 5.50 (1H, t, *J* = 6.1 Hz), 5.85 (1H, d, *J* = 3.4 Hz), 6.08 (1H, dd, *J* = 5.6, 2.0 Hz); ^13^C NMR (CDCl_3_, 100 MHz) δ 26.1, 26.7, 39.1, 74.0, 74.5, 80.3, 99.5, 105.5, 111.3, 202.1; HRMS (ESI-orbitrap) *m*/*z*: [M + Na]^+^; calculated for C_10_H_13_O_3_^79^BrNa 282.9940; Found 282.9940.

#### 3.2.11. Unsaturated Ester (1*R*)-**13**

To a solution of (1*R*)-**12** (349 mg, 1.34 mmol) in CH_3_CN-H_2_O (5.0 mL, 10/1) was added TsOH-H_2_O (35.0 mg) at room temperature under Ar atmosphere. The mixture was stirred at 60 °C for 6 h, quenched with saturated NaHCO_3_, extracted with AcOEt, washed with brine, dried over Na_2_SO_4_, filtered, and concentrated in vacuo. Crude hemiacetal was directly employed in the next reaction.

To a solution of the crude hemiacetal in CH_2_Cl_2_ (5.0 mL) was added *n*Bu_3_P=CHCO_2_Et (1.00 M in toluene, 2.76 mL, 2.76 mmol) at room temperature under Ar atmosphere. The mixture was stirred for 1 h and concentrated in vacuo. The crude product was purified by silica gel column chromatography (AcOEt:hexane = 30:70) to give unsaturated ester (1*R*)-**13** (154 mg, 0.530 mmol, 40% over 2 steps) as a colorless oil: [α]_D_^23^ −48.4 (c 0.50, CHCl_3_); IR (neat) 3851, 3735, 3649, 1698, 1540, 1181, 1034, 772 cm^–1^; ^1^H NMR (400 MHz, CDCl_3_) δ 1.30 (3H, t, *J* = 7.1 Hz), 1.76–1.91 (2H, m), 4.20 (2H, q, *J* = 7.2 Hz), 4.60–4.64 (2H, m), 5.50 (1H, t, *J* = 5.6 Hz), 6.10 (1H, dd, *J* = 15.9, 2.0 Hz), 6.14 (1H, dd, *J* = 5.6, 2.0 Hz), 6.92 (1H, dd, *J* = 15.6, 4.9 Hz); ^13^C NMR (CDCl_3_, 100 MHz) δ 14.5, 43.3, 60.9, 69.1, 71.1, 75.3, 103.7, 121.0, 148.8, 166.7, 201.0; HRMS (ESI-orbitrap) *m*/*z*: [M + Na]^+^; calculated for C_11_H_15_O_4_^79^BrNa 313.0046; Found 313.0046.

#### 3.2.12. Bromoether (1*R*)-**14**

To a solution of (1*R*)-**13** (134 mg, 0.460 mmol) in CH_2_Cl_2_ (3.0 mL) was added DIBAL (1.00 M in toluene, 2.30 mL, 2.30 mmol) at 0 °C under Ar atmosphere. The mixture was stirred for 30 min, quenched with 1.00 M HCl, stirred for 30 min, extracted with AcOEt, washed with brine, dried over Na_2_SO_4_, filtered, and concentrated in vacuo to give crude alcohol, which was directly employed in the next reaction.

To a solution of the crude alcohol in CH_3_CN (5.0 mL) was added NBS (81.9 mg, 0.460 mmol) at 0 °C under Ar atmosphere. The mixture was stirred for 30 min, quenched with 20% Na_2_S_2_O_3_ aqueous solution, extracted with AcOEt, washed with brine, dried over Na_2_SO_4_, filtered, and concentrated in vacuo. The crude product was purified by silica gel column chromatography (AcOEt:hexane = 20:80) to give bromoether (1*R*)-**14** (76.0 mg, 0.230 mmol, 51% over 2 steps) as a colorless oil: [α]_D_^23^ −80.7 (c 0.52, CHCl_3_); IR (neat) 3900, 3851, 3735, 3647, 2931, 1698, 1540, 1457, 1038, 670 cm^–1^; ^1^H NMR (400 MHz, CDCl_3_) δ 2.04 (1H, ddd, *J* = 15.1, 6.1, 4.9 Hz), 2.25 (1H, dd, *J* = 13.4, 6.3 Hz), 3.98 (1H, dd, *J* = 12.1, 4.4 Hz), 4.04 (1H, dd, *J* = 12.2, 4.9 Hz), 4.15–4.24 (2H, m), 4.60 (1H, t, *J* = 3.9 Hz), 4.91–4.97 (1H, m), 5.44 (1H, t, *J* = 5.9 Hz), 6.08 (1H, dd, *J* = 5.6, 2.0 Hz); ^13^C NMR (CDCl_3_, 100 MHz) δ 30.9, 40.5, 51.1, 65.8, 74.2, 75.6, 84.1, 101.4, 201.4; HRMS (ESI-orbitrap) *m*/*z*: [M + Na]^+^; calculated for C_9_H_12_O_3_^79^Br_2_Na 348.9045; Found 348.9048.

#### 3.2.13. TBS Ether (1*R*)-**15**

To a solution of (1*R*)-**14** (61.5 mg, 0.190 mmol) in CH_2_Cl_2_ (1.0 mL) were added 2,6-lutidine (110 μL, 0.950 mmol) and TBSOTf (131 μL, 0.570 mmol) at 0 °C under Ar atmosphere. The mixture was stirred at room temperature for 2 h, quenched with saturated NaHCO_3_, extracted with AcOEt, washed with 1.00 M HCl and brine, dried over Na_2_SO_4_, filtered, and concentrated in vacuo. Crude TBS ether (1*R*)-**15** was sufficiently pure without column chromatography: [α]_D_^23^ −80.9 (c 0.23, CHCl_3_); IR (neat) 3903, 3733, 2926, 1714, 1507, 1258, 1118, 836, 670 cm^–1^; ^1^H NMR (400 MHz, CDCl_3_) δ 0.07 (3H, s), 0.08 (3H, s), 0.12 (3H, s), 0.16 (3H, s), 0.91 (18H, s), 1.91 (1H, ddd, *J* = 13.2, 10.2, 3.9 Hz), 2.10 (1H, dd, *J* = 13.2, 5.9 Hz), 3.97–4.10 (4H, m), 4.50 (1H, d, *J* = 3.4 Hz), 4.79–4.85 (1H, m), 5.45 (1H, t, *J* = 6.1 Hz), 6.04 (1H, dd, *J* = 5.9, 2.0 Hz); ^13^C NMR (CDCl_3_, 100 MHz) δ −5.6, −5.1, −4.7, 0.7, 17.7, 18.1, 25.5, 25.6, 29.4, 41.4, 52.2, 64.7, 64.8, 73.0, 73.4, 75.1, 76.9, 82.6, 82.6, 101.8, 201.1; HRMS (ESI-orbitrap) *m*/*z*: [M + Na]^+^; calculated for C_21_H_40_O_3_^79^Br_2_NaSi_2_ 577.0775; Found 577.0779.

#### 3.2.14. Alcohol (1*R*)-**8**

To a solution of the crude (1*R*)-**15** in CH_2_Cl_2_-MeOH (0.20 mL, 2:1) was added CSA (13.2 mg, 0.0570 mmol) at room temperature under Ar atmosphere. The mixture was stirred for 90 min, quenched with Et_3_N (7.90 μL, 0.0570 mmol), and concentrated in vacuo. The crude product was purified by silica gel column chromatography (AcOEt:hexane = 30:70) to give alcohol (1*R*)-**8** (42.6 mg, 0.0969 mmol, 51% over 2 steps) as a colorless oil: [α]_D_^23^ −129.1 (c 0.30, CHCl_3_); IR (neat) 3447, 3057, 2954, 2929, 2884, 2856, 2362, 1959, 1474, 1437, 1361, 1255, 1178, 1108, 1044, 936, 836, 807, 776, 710 cm^–1^; ^1^H NMR (400 MHz, CDCl_3_) δ 0.13 (3H, s), 0.17 (3H, s), 0.92 (9H, s), 1.87 (1H, m), 2.11 (1H, dd, *J* = 13.1, 5.8 Hz), 2.62 (1H, dd, *J* = 8.0, 5.3 Hz), 3.95–4.01 (2H, m), 4.12 (1H, dd, *J* = 10.0, 2.4 Hz), 4.50–4.52 (1H, m), 4.83–4.89 (1H, m), 5.45 (1H, t, *J* = 5.8 Hz), 6.06 (1H, t, *J* = 5.6 Hz); ^13^C NMR (CDCl_3_, 100 MHz) δ −4.5, −4.0, 18.3, 26.1, 41.8, 50.8, 66.2, 73.6, 74.3, 76.0, 76.4, 85.7, 101.7, 201.8; HRMS (ESI-orbitrap) *m*/*z*: [M + Na]^+^; calculated for C_15_H_26_O_3_^79^Br_2_NaSi 462.9910; Found 462.9914.

### 3.3. CD Calculations

Theoretical CD spectra were obtained by following a typical calculation procedure [34] as described in the following. The initial structure was constructed on a graphical user interface (GaussView) considering the absolute configuration of interest and subjected to a conformational search with CONFLEX 8 [39,40] using MMFF94S (2010-12-04HG) as the force field, in which the initial stable conformers were generated for up to 50 kcal/mol. Confirming that the absolute configuration in focus was retained, stable conformers with >1% of abundance were further optimized by the DFT method supposing acetonitrile as the solvent (polarizable continuum models: PCM) at the approximation level of double-zeta with the hybrid functional B3LYP and basis set 6-31G(d). The conformers obtained were analyzed their populations with their Boltzmann distribution. Based on their energies that were obtained from the internal energies and vibrational corrections, the stable conformers that covered >90% of abundance as summation were estimated (two conformers for (1*S*)-**12**, two for (1*R*)-**12**, five for (1*S*)-**13**, six for (1*R*)-**13**, one for (1*S*)-**14**, one for (1*R*)-**14**, one for (1*S*)-**15**, and one for (1*R*)-**15**). All the relevant conformers were subjected to time-dependent simulations (TD-DFT) at the level of double-zeta approximation using cc-pVDZ basis set and hybrid functional B3LYP in acetonitrile (PCM method). The resultant rotational strengths were converted into Gaussian curves (bandwidth sigma = 3000 cm^–1^) and summed to give the theoretical CD spectrum for each compound. The best match between the experimental and the theoretical ECD curves through all the compounds was obtained when B3LYP/cc-pVDZ was applied.

## Data Availability

The data presented in this study are available on request from the corresponding author.

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
