# Peer review of "Assignment of Absolute Configuration of Bromoallenes by Vacuum-Ultraviolet Circular Dichroism (VUVCD)"

_molecules, 2021, doi:10.3390/molecules26051296_

Round 1

Reviewer 1 Report

The topic of the study concerns the natural products containing bromoallene moieties which possess axial chirality. In particular, the Lowe’s rule allows the determination of their absolute configuration through the specific rotation of the bromoallene functionality showing a large absolute value. Nevertheless, this empirical rule is not necessarily obeyed when other stereogenic centers are present in the molecule, therefore a more reliable method is required. The authors report here on the determination of the absolute configuration of bromoallenes on the basis of VUVCD spectroscopy. A series of diastereoisomeric pairs of bromoallenes, prepared by modifying the synthetic scheme of omaezallene, has been prepared and investigated. The synthetic protocols are clearly explained and appropriate experimental details are provided. The VUVCD spectra clearly showed positive or negative Cotton effects around 180-190 nm according to the configuration of the bromoallene employed, revealing the potential of VUVCD for determining absolute stereochemistry. DFT calculations support the experimental findings. In particular, the signs of the Cotton effect in the bromoallenes are in agreement with the absolute configurations in both experimental VUVCD and theoretical CD spectra for all the models of this study. These results demonstrate that the VUVCD method is very helpful for the unambiguous assignment of the absolute configuration of bromoallenes. The work is very solid, nicely presented and of interest for researchers in the fields of natural products and chirality in the broadest sense. I support its publication.

Author Response

The topic of the study concerns the natural products containing bromoallene moieties which possess axial chirality. In particular, the Lowe’s rule allows the determination of their absolute configuration through the specific rotation of the bromoallene functionality showing a large absolute value. Nevertheless, this empirical rule is not necessarily obeyed when other stereogenic centers are present in the molecule, therefore a more reliable method is required. The authors report here on the determination of the absolute configuration of bromoallenes on the basis of VUVCD spectroscopy. A series of diastereoisomeric pairs of bromoallenes, prepared by modifying the synthetic scheme of omaezallene, has been prepared and investigated. The synthetic protocols are clearly explained and appropriate experimental details are provided. The VUVCD spectra clearly showed positive or negative Cotton effects around 180-190 nm according to the configuration of the bromoallene employed, revealing the potential of VUVCD for determining absolute stereochemistry. DFT calculations support the experimental findings. In particular, the signs of the Cotton effect in the bromoallenes are in agreement with the absolute configurations in both experimental VUVCD and theoretical CD spectra for all the models of this study. These results demonstrate that the VUVCD method is very helpful for the unambiguous assignment of the absolute configuration of bromoallenes. The work is very solid, nicely presented and of interest for researchers in the fields of natural products and chirality in the broadest sense. I support its publication.

>>> We are grateful for the precise and sincere comments.

Reviewer 2 Report

In this work, the authors synthesized diasteomeric chiral bromoallenes and used vacuum ultraviolet-circular dichroism (VUVCD) as a tool to determine the absolute configuration of the allene moieties. Their study revealed that VUVCD together with theoretical CD spectra calculation can indeed be successfully employed as such, thus representing a valid alternative to the empiric Lowe’s rule, which has been used so far. The work is clear, well presented and supported by the experimental data. In my opinion, it may be considered for publication after minor revisions.

More in details, the authors should reference in the introductive part the work by Elsevier and co-workers (JACS, 1985, 107, 2537-2547), just to clearly indicate that VUVCD has already been recorded previously for chiral allenes, including bromoallenes (even if that work was not aimed at presenting a VUVCD-based method for the determination of the absolute configuration of chiral allenes).

At the beginning of chapter 2 (“Results and discussion”) the authors write: “construction of propargyl alcohol 5, a stereospecific precursor for the bromoallene, was previously carried out in a stereoselective manner with acetylide and hemiacetal 4 obtained from D-glucose through aldehyde 3 (Scheme 1). We believe this high stereoselectivity was due to the hydroxy group in the α-hydroxyaldehyde, which was in equilibrium with hemiacetal 4.” I think this part is referred to a previously published work, so I suggest adding the right citation. In addition, the first row of Scheme 1 may be improved, as it seems not so clear to me. For example, the equilibrium between the hemiacetal and aldehyde may be drawn and a drawing for the R substituent may be included.

The authors assigned the absolute configuration of esters 9 and 10 by using the Mosher method: it would be nice either to have the principal data included in the main text or to have a reference to the appropriate figures and tables of the supplementary information. A citation to the right reference is also needed after the sentence: “1H and 13C NMR spectra of (1S)-8 were identical with those of the previous intermediate of omaezallene”.

The author should explain better the following sentence: “The signs of the CD spectra of all of these compounds agreed well with the reported ones [33]”. In the cited work the CD analysis was not carried out on the compounds investigated by the authors of the manuscript, but they were done on other chiral allenes. I think that a more correct form may be "the relationship between the signs of the CD spectra of all these compounds and their absolute configuration is in line with that obtained for other chiral allenes" (or something like that).

Since the authors are claiming a possible general method to determine the absolute configuration of bromoallenes and also its feasibility in solution (“while these transitions were previously either non-observable or visualized only by gas-phase measurements, our results demonstrate the potential of VUVCD for the robust assignment of absolute configurations of bromoallenes”), a more detailed setup of the instrumentation and experiments should be included (instrumentation, parameters, concentration, solvents, and so on) and discussed.

I understand that there is an ongoing work on other chiral allenes and I strongly believe that the combination of CD measurements and theoretical calculation of the spectra is a more robust method respect to the Lowe’s rule. However, basing the claim that the VUVCD can be used “instead of Lowe’s rule because no exceptions were observed in this study” is not right. Indeed, “no exceptions were observed in this study" not only for the VUVCD measurements but also by using Lowe's rule. Therefore, I would just omit the last part of the sentence (“because no exceptions were observed in this study”). Otherwise, the authors will need to include at least an example of a bromoallene compound that does not follow the Lowe’s rule but present the “correct” CD sign.

Author Response

Comments and revisions for reviewer 2:

In this work, the authors synthesized diasteomeric chiral bromoallenes and used vacuum ultraviolet-circular dichroism (VUVCD) as a tool to determine the absolute configuration of the allene moieties. Their study revealed that VUVCD together with theoretical CD spectra calculation can indeed be successfully employed as such, thus representing a valid alternative to the empiric Lowe’s rule, which has been used so far. The work is clear, well presented and supported by the experimental data. In my opinion, it may be considered for publication after minor revisions.

More in details, the authors should reference in the introductive part the work by Elsevier and co-workers (JACS, 1985, 107, 2537-2547), just to clearly indicate that VUVCD has already been recorded previously for chiral allenes, including bromoallenes (even if that work was not aimed at presenting a VUVCD-based method for the determination of the absolute configuration of chiral allenes).

At the beginning of chapter 2 (“Results and discussion”) the authors write: “construction of propargyl alcohol 5, a stereospecific precursor for the bromoallene, was previously carried out in a stereoselective manner with acetylide and hemiacetal 4 obtained from D-glucose through aldehyde 3 (Scheme 1). We believe this high stereoselectivity was due to the hydroxy group in the α-hydroxyaldehyde, which was in equilibrium with hemiacetal 4.” I think this part is referred to a previously published work, so I suggest adding the right citation. In addition, the first row of Scheme 1 may be improved, as it seems not so clear to me. For example, the equilibrium between the hemiacetal and aldehyde may be drawn and a drawing for the R substituent may be included.

>>> We are grateful for the careful reading by the reviewer. We have corrected the scheme 1 in which whole structures of hemiacetal 4’ and propargyl alcohol 5 are added (line 57). The citation for this part was also added (line 53).

The authors assigned the absolute configuration of esters 9 and 10 by using the Mosher method: it would be nice either to have the principal data included in the main text or to have a reference to the appropriate figures and tables of the supplementary information.

>>> We are grateful for the suggestion by the reviewer. For the careful attention to reader, ‘(see Figure S1 in the Supplementary Material)’ have been added (line 64). Supplementary Material was also improved.

A citation to the right reference is also needed after the sentence: “1H and 13C NMR spectra of (1S)-8 were identical with those of the previous intermediate of omaezallene”.

>>> According to the suggestion by the reviewer. The citation for this part has been added (line 72).

The author should explain better the following sentence: “The signs of the CD spectra of all of these compounds agreed well with the reported ones [33]”. In the cited work the CD analysis was not carried out on the compounds investigated by the authors of the manuscript, but they were done on other chiral allenes. I think that a more correct form may be "the relationship between the signs of the CD spectra of all these compounds and their absolute configuration is in line with that obtained for other chiral allenes" (or something like that).

>>> We are grateful for the careful reading by the reviewer. We corrected the corresponding sentence to be “The relationship between the signs of the CD spectra of all of these compounds and their absolute configurations is consistent well with that obtained for other chiral allenes” (lines 83-84).

Since the authors are claiming a possible general method to determine the absolute configuration of bromoallenes and also its feasibility in solution (“while these transitions were previously either non-observable or visualized only by gas-phase measurements, our results demonstrate the potential of VUVCD for the robust assignment of absolute configurations of bromoallenes”), a more detailed setup of the instrumentation and experiments should be included (instrumentation, parameters, concentration, solvents, and so on) and discussed.

>>> We appreciate for the reviewer’s suggestion. The explanation of instrumental setups and conditions for VUVCD measurements was added more in detail as follows: “The VUVCD spectra were measured in the wavelength region from 200 to 175 nm at 25°C using a VUVCD spectrophotometer at the Hiroshima Synchrotron Radiation Center (HiSOR), Hiroshima University [18]. This spectrophotometer uses a synchrotron radiation light source to observe the CD spectrum in the high energy region. The details of the optical devices of the spectrophotometer are available elsewhere [15, 16]. The VUVCD measurements were carried out using an assembled-type optical cell with CaF2windows [18]. The path length in the cell was adjusted using a Teflon spacer to 50 mm. The sample solutions were freshly prepared by dissolving in acetonitrile at an appropriate concentration, which was determined based on the reported molar absorption coefficient [33]. All of the spectra were recorded using a 1.0-mm slit, 4-s time constant, 20-nm/min scan speed, and four accumulations.” (lines 133-140)

I understand that there is an ongoing work on other chiral allenes and I strongly believe that the combination of CD measurements and theoretical calculation of the spectra is a more robust method respect to the Lowe’s rule. However, basing the claim that the VUVCD can be used “instead of Lowe’s rule because no exceptions were observed in this study” is not right. Indeed, “no exceptions were observed in this study" not only for the VUVCD measurements but also by using Lowe's rule. Therefore, I would just omit the last part of the sentence (“because no exceptions were observed in this study”). Otherwise, the authors will need to include at least an example of a bromoallene compound that does not follow the Lowe’s rule but present the “correct” CD sign. After checking the revision of this paper by the authors, I consider that some of my previous concerns are still not properly addressed. Authors try to demonstrate the absolute configuration of a natural compound by comparison of its properties with those of synthetic samples of the same molecule. The only link between the natural and the synthetic samples seem to be the comparison of a chiral HPLC analysis in which, as stated in my previous review, a simple confusion in the identification of the synthetic samples during this analysis might lead to opposite results.

>>> We agreed with the reviewer’s point and omitted the specified part of sentence (lines 118-119).